# Comparison of Osmotic Resistance, Shape and Transmembrane Potential of Erythrocytes Collected from Healthy and Fed with High Fat-Carbohydrates Diet (HF-CD) Pigs—Protective Effect of *Cistus incanus* L. Extracts

**DOI:** 10.3390/ma14041050

**Published:** 2021-02-23

**Authors:** Sylwia Cyboran-Mikołajczyk, Robert Pasławski, Urszula Pasławska, Kacper Nowak, Michał Płóciennik, Katarzyna Męczarska, Jan Oszmiański, Dorota Bonarska-Kujawa, Paweł Kowalczyk, Magdalena Wawrzyńska

**Affiliations:** 1Department of Physics and Biophysics, Wrocław University of Environmental and Life Sciences, Norwida 25, 50-375 Wrocław, Poland; sylwia.cyboran@upwr.edu.pl (S.C.-M.); katarzyna.meczarska@upwr.edu.pl (K.M.); dorota.bonarska-kujawa@upwr.edu.pl (D.B.-K.); 2Department of Veterinary Surgery, Veterinary Insitute Nicolaus Copernicus University in Toruń, Poland, ul. Gagarina 7, 87-100 Toruń, Poland; r.paslawski@umk.pl; 3Department of Diagnostics and Clinical Sciences, Veterinary Insitute Nicolaus Copernicus University in Toruń, Poland, ul. Gagarina 7, 87-100 Toruń, Poland; urszula.paslawska@umk.pl; 4Department of Internal Diseases and Clinic of Diseases of Horses, Dogs and Cats, Faculty of Veterinary Medicine, Wroclaw University of Environmental and Life Sciences, Grunwaldzki Sq. 47, 50-366 Wroclaw, Poland; kacper.nowak@upwr.edu.pl; 5Department of Biochemistry and Molecular Biology, Faculty of Veterinary Medicine, Wroclaw University of Environmental and Life Sciences, Norwida 31 Str., 50-375 Wroclaw, Poland; michal.plociennik@upwr.edu.pl; 6Department of Fruit, Vegetable and Plant Nutraceutical Technology, Wrocław University of Environmental and Life Sciences, Norwida 25, 50-375 Wrocław, Poland; jan.oszmianski@upwr.edu.pl; 7Department of Animal Nutrition, The Kielanowski Institute of Animal Physiology and Nutrition, Polish Academy of Sciences, 05-110 Jabłonna, Poland; 8Department of Preclinical Studies, Faculty of Health Sciences, Wrocław Medical University, Wybrzeże Paustera 1, 50-367 Wrocław, Poland; magdalena.wawrzynska@umed.wroc.pl

**Keywords:** erythrocytes, fat-carbohydrates diet (HF-CD), *Cistus incanus* L. extracts, pigs, osmotic resistance

## Abstract

Long-term high fat-carbohydrates diet (HF-CD) contributes to the formation of irreversible changes in the organism that lead to the emergence of civilization diseases. In this study, the impact of three-month high-fat diet on the physical properties of erythrocytes (RBCs) was studied. Furthermore, the biological activity of *Cistus incanus* L. extracts, plant known with high pro-health potential, in relation to normal and HF-CD RBCs, was determined. Obtained results have shown that, applied HF-CD modified shape, membrane potential and osmotic resistance of erythrocytes causing changes in membrane lipid composition and the distribution of lipids. The impact of HF-CD on physical properties of RBCs along with atherosclerotic lesions of the artery was visible, despite the lack of statistically significant changes in blood morphology and plasma lipid profile. This suggests that erythrocytes may be good markers of obesity-related diseases. The studies of biological activity of *Cistus incanus* L. extracts have demonstrated that they may ameliorate the effect of HF-CD on erythrocytes through the membrane-modifying and antioxidant activity.

## 1. Introduction

A diet enriched with saturated fatty acids and carbohydrates (high fat-carbohydrates diet, HF-CD) consequently leads to reversible, than irreversible metabolic disturbances, which result in the emergence of civilization diseases, including obesity and accompanying diseases such as atherosclerosis and diabetes [1]. HF-CD induces systemic inflammation, what is related to the disorder of oxidation-reduction balance and the excess of free radicals damaging to basic cell components [2]. The erythrocytes circulation is particularly exposed to the free radicals due to their transporting functions in delivering oxygen to tissues. They are widely recognized for their vital role in transporting and delivering oxygen to the tissues. Furthermore, RBCs regulate blood vessel function by releasing adenosine triphosphate (ATP) and the levels of key circulating chemokines, modulating nitric oxide (NO)-dependent vasorelaxation [3,4]. The RBCs are important in the progression and instability of atherosclerotic plaque. They can accumulate into the atherosclerotic plaque after a plaque rupture or erosion and will increase the cholesterol content as well as the inflammatory/oxidative stress of the plaque [5]. Thus, the physical properties of the RBC membrane, and therefore its cholesterol content, are of great importance in the development of atherosclerosis. Gold and Phillips in 1990 and Varshney 2019 have shown that the level of cholesterol in RBC membrane is independent of acute changes in plasma cholesterol [6,7]. Literature data indicate that the effects of HF-CD on RBCs depend on the length and composition of the diet and the research model used, therefore, the results obtained in various studies are often inconsistent. It was shown that hypolipidemic diet does not influence the lipid composition of the RBC membrane [8,9]. Unruh et al. have shown that HFD induces the dysfunction of RBCs in mice [10]. Other studies also indicate HF-CD induced changes in RBC membrane i.e., changes in osmotic resistance or membrane elasticity [11]. We investigated whether HF-CD may induce the dysfunction of RBCs in pigs that were fed for three month with proatherosclerotic diet enriched with fat and carbohydrates. The degree of hemolysis of HF-CD RBCs, their osmotic resistance and shapes were investigated and compared with control cells. Furthermore, the impact of two extracts of *Cistus incanus* L. on physical properties of RBCs membrane was investigated in order to determine their biological activity and potential ability to ameliorate the physiological and physical function of RBC obtained from HF-CD pigs. Cistus, as a medicinal plant, was already used in natural medicine in ancient times. It phenolic extracts show high antioxidant activity [12,13,14], because contain the large amount of flavonoids that are natural scavengers of (ROS), responsible for the oxidation of biological systems. Flavonoids may regulate and support the treatment of obesity-related diseases by modulating fat metabolism [15,16]. Apart from the high antioxidant potential phenolic extracts of Cistus originating from various species, in relation to organisms, also show a number of pro-health properties. They are i.e., anti-inflammatory, antiviral, antibacterial, and antifungal [17,18,19,20]. In recent years, were conducted an intensive research on the phenolic composition of individual cistus species and their biological activity. Despite it, the effects of the cistus extracts on the organism at the cellular level, and the specific molecular mechanism responsible for the effects are still not fully explained. Therefore, in this work were carried out an innovative biophysical research, including the effects of two polyphenolic extracts from cistus on physical properties of RBCs, which play a key role in the organism, and determine these effects. The basis on the hemolytic and antioxidant and membrane potential, as well as on the shape of RBCs, the biological activity of *Cistus incanus* L. extracts were determined. The studies were conducted in relation to RBCs obtained from pigs fed with balanced diet and diet enriched in saturated fat and carbohydrates (imitating the so-called “western diet”).

## 2. Materials and Methods

### 2.1. Animals

The experiment was performed with the approval No. 23/2009 of the 1st Local Ethics Committee for Animal Experimentation of the Institute of Immunology and Experimental Therapy in Wroclaw, Poland. We used a swine model of atherosclerosis described earlier [1]. Namely, all pigs (polish white domestic pig, *suis scrofa*) were housed in a single room, with a temperature of 18–20 °C and 60–75% humidity. Nine pigs were fed by three months the unbalanced diet having a 3200 kcal/kg energy content. This diet (HF-CD) [1]. Second—control group (eleven pigs) had limited access to food proportionally to the body weight with daily maximum calorie intake up to 4200 kcal/pig. Before the examination swine fasted 12 h, with full access to the water. Pigs were premedicated with an intramuscular injection of 1 mg/m^2^ body surface area (BSA) of medetomidine hydrochloride (Cepetor, CP-Pharma, Germany), 5 mg/m^2^ BSA midazolam (Midanium, WZF Polfa, Poland) and 264 mg/m^2^ BSA ketamine (Bioketan, Vetoquinol Biowet, Poland) and then intubated [21]. An ear vein was punctured to collect a blood sample.

The hematological examination (WBC, RBC, HGB, HCT, MCV, PLT) was performed by hematology analyzer IDEXX X LaserCyte (Japan) and a Horiba ABC animal blood counter (USA). The biochemical examinations of Na, K, Ca^2+^, Mg, Fe, glucose, urea, creatinine, AspAT, ALT, CRP, TG, total cholesterol HDL, LDL were performed in Wroclaw University of Environmental and Life Sciences (Table 1).

After the end of experiment, samples of miocardium (possess left descendens coronary artery) were obtained for cyto and histopathological analysis with haematoxylin and eosin staining. Histological specimens were evaluated using an Olympus BX41 light microscope with a built-in Olympus Color View IIIu U-CMAD3 camera. Atherosclerotic lesions in the vessel wall were classified according to the histological structure using an eight-point scale [Stary, 2000]: I°—isolated macrophage foam cells, II°—the formation of multiple cell layers, III°—the formation of isolated extracellular lipid core IV°—the formation of a confluent extracellular lipid core, V°—the formation of fibromuscular tissue layers, VI°—surface defect, hematoma, thrombosis, VII°—a predominance of calcification, VIII°—a predominance of fibrous changes.

We do not observe statistically significant differences in blood morphological and biochemical parameters in samples taken after 3 months of pigs feeding HF-CD, although an obvious trend towards total LDL cholesterol, glucose and HDL was quite visible Histopathological examination revealed atherosclerosis of the 1st and 2nd degree. In the middle (tunica media) and inner (tunica intima) layers of the artery were observed areas of local thickening (Figure 1B).

### 2.2. Extracts

Two extracts from *Cistus incanus* L. (EC1 and EC2) were used. The raw material the gray purge herb *Cistus incanus* L. for the preparation of extracts came from Albania. The material immediately after collection was lyophilized for 24 h (Alpha 1-4 LSC, Martin Christ Gefriertrocknungsanlagen GmbH, Germany). Phenolic compounds isolated form the lyophilized powder using two methods. The EC1 extract was obtained by extraction with 50% methanol (the procedure was described in Gąsiorowski [22] the ratio of this solvent to the raw material was 3:1 (*v*/*v*). EC2 extract was prepared according to the method described earlier in [23]. In this method ethyl acetate extraction was used, which causes selective precipitation of catechin and procyanidin groups [24,25,26,27,28,29,30,31,32,33,34,35,36,37,38,39,40]. Polyphenols contained in the extracts were added to the Table 2.

The dominant component of both extracts is myricetin 3-rhamnoside (M3R), whose content is 31% in EC1 and 54% in EC2 of all phenolic components, respectively.

### 2.3. Reagents

Oxidation inducer 2,2′-azobis(2-amidinopropane) dihydrochloride (AAPH), 3,3′-dipropylthiadicarbocyanine iodide (DiSC3(5)), valinomycin and myricetin 3-rhamnoside (M3R) were purchased from Sigma-Aldrich, Inc., Steinheim, Germany. All other reagents were analytically pure.

## 3. Methods

### 3.1. The Degree of Hemolysis RBCs

The degree of hemolysis of RBCs obtained from pigs fed a balanced diet and HF-CD and the ability of the extracts to induce hemolysis of red blood cells were studied spectrophotometrically [41].

### 3.2. Osmotic Resistance of Erythrocytes

Osmotic resistance assay was performed according to [42].

### 3.3. Erythrocytes Transmembrane Potential

Erythrocytes transmembrane potential was performed according to [39].

### 3.4. Shape of Erythrocytes

*Shape of erythrocytes* was performed according to [25,26,39,43].

### 3.5. Oxidation of RBCs Induced by AAPH

*Oxidation of RBCs induced by AAPH* was performed according to [39,44].

## 4. Results

### 4.1. The Degree of Hemolysis of HF-CD RBCs

The calculated percentage degree of hemolysis of control and HF-CD RBCs is 2.6 ± 1.2% and 2.9 ± 1.4%, respectively. The results indicate no differences between hemolysis of HF-CD RBCs relative to hemolysis of control RBCs. It means that HF-CD do not induces increases hemolysis of RBCs.

### 4.2. Hemolytic Activity of EC1 and EC2 Extracts

The hemolytic activity of EC1, EC2 and M3R were determined using UV-Vis spectroscopy. The results have shown that RBCs treated with used compounds do not show increased hemolysis relative to control cells to the concentration of 0.1 mg/mL (the results are not presented). It means that they do not damage both health and HF-CD membrane of RBCs and can be safely used in a wide range of concentrations.

### 4.3. The Impact of HF-CD on Osmotic Resistance of Erythrocytes

The impact of HF-CD on osmotic resistance of RBCs was determined. Relationships between the percentage of hemolysis of RBCs and NaCl concentrations, i.e., hemolytic curves are shown in Figure 2. A comparison of control RBCs with cells collected from pigs fed HF-CD (Figure 2) showed that these RBCs possess increased resistance to changes in osmotic pressure of surrounding environment. It means that they are more resistant to changes in osmotic pressure.

### 4.4. The Impact of EC1 and EC2 on Osmotic Resistance of Healthy and HF-CD RBCs

The results presented in Figure 3B,C indicate that EC1 and EC2 extracts used at 0.05 mg/mL induces changes in osmotic resistance of both healthy and HF-CD erythrocytes. They caused a shift of the osmotic resistance curve towards lower NaCl concentrations to varying degrees. For erythrocytes collected from healthy pigs only EC1 extract increases their osmotic resistance (Figure 3B). The EC2 extract practically do not induce changes, no statistically significant differences between EC2-treated and control cells were found. In the case of HF-CD RBCs, the increased osmotic resistance was observed for both extract (Figure 3C). However, the changes induced by EC1 were comparable to those observed for EC2 extract. Furthermore, studies carried out for the main component of both extracts, i.e., M3R, do not show changes in osmotic resistance of erythrocytes modified with this compound in relation to the control RBCs (results are not presented).

### 4.5. Transmembrane Potential of RBCs and HF-CD RBCs

The fluorimetric studies using the DiSC3(5) probe have allowed to determine the membrane potential of RBCs collected from healthy pigs and those fed HF-CD. The calculated values of transmembrane potential are given in Table 3.

The calculated results presented in Table 3 indicate that the transmembrane potential of HF-CD RBCs slightly differ from the transmembrane potential of RBCs collected from healthy pigs. There is visible a slight growing trend in HF-CD RBCs i.e., the potential assumes lower negative value.

### 4.6. Transmembrane Potential of Erythrocytes Modified by EC1 and EC2 Extract

The modification of RBCs with EC1 and EC2 extracts results in changes on transmembrane potential (Table 3). In the case of healthy RBCs, both extracts decrease membrane potential in the same extent, which reaches lower negative values. Similar slight decreases but not statistically significant were observed in the case of HF-CD RBCs treated by the extracts, in particular under the influence of EC2. Thus, the presence of the EC2 extract molecules causes that the electrical properties of HF-CD RBCs are slightly closer to those observed for healthy RBCs. Furthermore, studies on the effect of M3R on the transmembrane potential of RBCs did not show any changes in comparison to control cells.

### 4.7. Shapes of HF-CD RBCs

The imagines registered using optical microscope showed changes in the shape of RBCs obtained from HF-CD pigs compared to control pigs. The percentage share of RBCs individual forms was calculated on the basis of photographs of at least 300 blood cells. The comparison of RBCs and HF-CD RBCs shapes is shown in the Figure 4A.

The reduced numbers of discocytes and increased number of irregularly shaped blood cells such as discostomatocytes and dyscoechinocytes have been observed in the case of HF-CD RBCs in comparison to control RBCs (4A).

### 4.8. Shapes of RBCs Modified by EC1 and EC2 Extracts

The modification of RBCs with EC1 and EC2 extracts induces formation of different forms of echinocytes (Figure 4B). The EC1 and EC2-modified RBCs transition from normal, biconcave discocytes, to distinct intermediate echinocytic stages: Mainly discoechinocytes (DE1) and echinocytes (E2), however, as results have shown, their ability to change the shape of RBCs is at a similar level. The effect of M3R is much smaller than those of EC1 and EC2; for M3R observed mainly the formation of first form of echinocytes (DE1). In the case of HF-CD RBCs (3C), the presence of extract molecules in RBC membrane reduce number of stomatocytes and significantly increase the number of echinocytes (DE1 and DE2). The share of individual cell shapes becomes more similar to that of control RBCs, however the increase in DE1 and E2 echinocytes by about 20% in both cases is still observed.

### 4.9. Protective Effect of EC1 and EC2 against AAPH-Induced Oxidation of RBCs

Used extracts do not have a destructive effect on the RBC membrane. The obtained results showed that the tested compounds show a high ability to scavenge free radicals, protecting RBCs against hemolysis caused by ROS. The concentrations of EC1 and EC2 extracts responsible for 50% inhibition of oxidative damage to RBCs are similar (Table 4), which suggests similar antioxidant activity of both extracts. Moreover, they showed that these extracts protect RBCs much better than their main component (M3R) used at the same concentration.

## 5. Discussion

Literature data unanimously indicate that the HF-CD is harmful and induces adverse changes in the organism, the progress of which depends on many factors. It may increase the concentration of saturated fatty acid in blood, expression of genes involved in inflammatory processes in adipose tissue, reduce insulin sensitivity and increase intrahepatic triglyceride content in humans [27,28]. In consequence, it contributes to the establishment of chronic inflammation and increased production and release of pro-inflammatory cytokines, notably TNF-α, IL-1 and IL-6, into the circulation [29]. The oxidation–reduction balance is disturbed and the number of free radicals increases.

It is very important to emphasize that the effects of HF-CD depend on animal models [30]; fat content of the diet time of exposition [31]. Our studies have shown that the effect of feeding pigs for 3 months with a diet enriched with fats and carbohydrates result in atherosclerotic lesions of blood vessels of 1st and 2nd degree. The HF-CD do not induce significant changes in hematological tests, glucose concentration and profile of fatty acids in the blood serum. However, a slight trend towards an increase in the level of total LDL cholesterol and glucose and a decrease in HDL was visible. Triglycerides and cholesterol reached significance in further observation as described elsewhere [1]. Therefore, the goal of the next stage of the studies was to determine whether HF-CD may affect the physical properties of RBCs. RBCs are good models of HF-CD impact because they readily exchange its free cholesterol with that of the serum lipoproteins; and they have virtually no capacity to synthesize and incorporate fatty acids into phospholipids. The impact of HF-CD on the degree of hemolysis, osmotic resistance, transmembrane potential, and the shape of RBCs was determined. Our studies indicate that the used HF-CD induces changes in the physical properties of RBC membranes, although its application does not induce significant changes in blood morphology and plasma lipid profile. The spectroscopic studies do not show increased hemolysis of RBCs obtained from HF-CD fed pigs, but they showed their increased resistance to changes in osmotic pressure. This is indicated by a shift in the hemolytic curve towards lower NaCl concentrations in comparison to control RBCs. It means that HF-CD RBCs are less able to release hemoglobin at varying osmotic pressure. Furthermore, as studies have shown, the transmembrane potential of HF-CD RBCs is significantly different from the transmembrane potential determined for blood cells from healthy pigs. The membrane potential of HF-CD RBCs is less negative potential value then control RBCs and it is associated with a slight increase in extracellular potassium ion concentration. It can therefore be presumed that in HF-CD pig, the permeation of potassium ions across the RBC membrane is slightly reduced. The shapes of RBCs are also changed due to HF-CD; the number of discocytes is significantly reduced and instead of them are formed mainly discostomatocytes and dyscoechniocytes. The observed changes in the physical properties of the membrane are probably caused by impaired lipid composition. It is highly likely that both vary: A membrane lipid composition and the distribution of lipids in both monolayers. Both the osmotic resistance of RBCs and the membrane potential indirectly depend on the lipid composition of the membrane, which in turn determines the fluidity of the membrane, and thus its ability to deform in narrow capillaries [2]. With a slight increase in cholesterol content in the RBC membrane, blood cells may show even greater resistance to changes in osmotic pressure [33]. With a significant increase in cholesterol content in the RBC membrane, its stiffness increases, and the deformability and susceptibility to lysis decrease, and thus their stability deteriorates significantly [34]. Moreover, the altered fluidity of the membrane modifies the activity of membrane proteins, and therefore its transport properties, and thus the membrane permeability to ions. In addition, during the breakdown of RBCs, hemoglobin is released, which, thanks to the presence of iron, generates ROS, which in turn increases the inflammation already present in the course of atherosclerosis [32,34]. Hagve et al. have shown the rise of osmotic resistance of rat RBCs fed a diet supplemented in n-3 fatty acid [9]. They do not observed changes in phospholipid or cholesterol content in the membrane but a small differences in distribution of the phospholipid subclass was visible (that the phosphatidylserine fraction was significantly increased and phosphatidylethanolamine decreased). These authors postulated that major lipid-related factors that may be involved in regulation of the physical state of the membrane are the phospholipid/cholesterol ratio, the fatty acid pattern of membrane phospholipids and the class distribution of phospholipids. Other authors also report increased osmotic resistance of rabbit RBCs incubated with cholesterol in vitro and a slight change in the red cell morphology with some spur cells appearing [33]. Furthermore, RBCs enriched with cholesterol may have ability to form echinocytes, while cholesterol-depleted cells may form stomatocytes [35]. The presence of additional cholesterol molecules in RBC membrane lead to changes in membrane fluidity, and in consequences the differences in the properties of outer leaflet relative to the inner leaflet. It may result in the formation of characteristic irregular shape of the cholesterol enriched RBCs [35]. This explanation is in accordance with the bilayer couple hypothesis [36] and predicts a change in the relative composition of the two leaflets will affect the overall shape of the cell membrane. Based on the obtained results we postulated that HF-CD may induced changes in membrane lipid composition and the distribution of lipids between monolayers. It consequences, it can lead to changes in membrane fluidity and/or the disturbances of ion permeability across the membrane [24]. The induced changes in the lipid environment of the membrane will affect the activity of membrane proteins, and thus the functioning of the entire cells. The diet we used in this study is rich in saturated fat and carbohydratessucrose, leading to obesity and changes in lipids profile, but the duration was not long enough to induce frank metabolic syndrome and large changes in the physicochemical properties of the RBC membrane. Therefore, there is a need to undertake additional studies with the prolongation of used HF-CD in order to confirm the obtained changes and checking that the observed changes will increase with the duration of the diet. Thus, as a study have shown the slight RBC changes, and analogous to endothelial dysfunction, may occur early in the course of diet-induced obesity and function as a marker and, possibly, mediator of atherosclerosis. In the next step of this study, the impact of two extracts of *Cistus incanus* L. (EC1 and EC2) on physical properties on healthy and HF-CD RBCs was determined. Firstly, it was checked whether the EC1 and EC2 could induce damage of the RBC membrane, by testing their hemolytic activity. It is known that disturbances in the transport of substances permeating through the membrane induce the differences in pressure on both sides of the RBCs. This, in effect, is responsible for the swelling of RBCs due to the water stream entering the cell and the outflow of hemoglobin as a result of membrane damage. As shown in this study, phenolic compounds contained in EC1 and EC2 extracts and M3R, in the concentration range studied, do not induce hemolysis. This means that they can be safely used in a wide range of concentrations. Next, the impact of the compounds on physical properties of RBCs was tested. The studies have shown that phenolic components of the extracts induce changes in RBCs membrane. In the case of control RBCs, EC1 and EC2 reduce osmotic fragility, making the membrane more resistant to hemoglobin release under the changes in osmotic pressure of the surrounding environment. The presence of extracts components in RBCs membrane, as studies have shown, also cause slight decrease of the value of transmembrane potential, that reaches greater negative values. Moreover, their interaction with RBC membrane, in consequence, induce changes in RBCs shape. According to bilayer couple hypothesis, used compounds bind mostly to the outer monolayer of RBCs membrane, because under their influence mainly different forms of echinocytes are formed [36]. Observed changes in osmotic stability, shape and transmembrane potential of RBCs are probably the results of binding of the extracts components mainly to the hydrophilic part of RBC membrane. It is possible, because extracts components such us glycosylated derivatives of myricetin, quercetin and kaempferol are rather hydrophilic. It is already known that hydrophilic flavonoids and oligomers interact with the area of head groups of lipids at the lipid–water interface, and this interaction may be associated with the formation of hydrogen bonds or have an electrostatic nature [37,38,39]. Their presence in this area may partially disrupts the interactions between lipid and/or protein and lipid molecules and consequently cause the observed changes in the physical properties of RBCs. Furthermore, the results indicate that the membrane-modifying activity of EC2 is much greater than EC1, which is related with the presence of a greater amount of identified small polyphenolic molecules. The activity of EC2 is also much higher than the activity of M3R the main component of the extract, the content of which is more than 50% of all identified ingredients. These results indicate that the mixture of polyphenolic compounds contained in *Cistus incanus* L., are much more effective than a main component used at the same concentration. Moreover, the differences in the activity of EC1 and EC2 extracts result from the method of obtaining polyphenolic compounds from plant material. In addition to the identified compounds, the EC1 polyphenol extract also contains others phenolic components with large molecular weight, which cannot be accurately identified by the chromatographic method used. The compounds contained in the extracts, beyond the ability to modify the properties of the RBCs membrane, effectively protect them from oxidative damage induced by AAPH. We do not observe any differences between antioxidant activity of EC1 and EC2, but their ability to scavenge free radicals is more effective than M3R. Our earlier studies using fluorimetric methods showed that the polyphenols contained in the EC1 and EC2 increase the lipid polar head motility in the hydrophilic part of the membrane and also slightly increase fluidity of the erythrocyte membrane. The observed changes in the physical properties of the erythrocyte membrane indicate that they are located mainly in the hydrophilic membrane area [40,43,44]. Taking into account the size and chemical structure of the extract’s ingredients it may be concluded that the slight changes in the hydrophobic area of the membrane are the results of changes in the hydrophilic area [40]. Extract’s components are rather not able to penetrate into the hydrophobic core, but they may electrostatically interact with polar head of lipids or form the hydrogen bonds with their OH groups. Their location on membrane-water interface, results in high protection against free radicals induced by the AAPH compound in aqueous medium. This location causes that they form a barrier on the surface of the membrane that limits the penetration of free radicals into the cell, protecting the cell components against oxidation.

In the case of HF-CD erythrocytes, both extracts of RBCs, making them more resistant to lysis in hypotonic solutions. They also slightly increase transmembrane potential, that value becomes closer to that of control RBCs. Furthermore, their interaction with HF-CD RBCs membrane results in a reduction of the number of stomatocytes and their conversion mainly into discoechinocytes, thus making the HF-CD RBCs shapes more similar to that of control RBCs. These results indicate that modification of RBCs (e.g., by injection into the blood stream) by *Cistus incanus* L. *extracts, may partially compensate the effect of HF-CD diet*. Therefore, in the future, additional research should be carried out in order to establish whether the diet enriched with *Cistus incanus* L. can protect dRBCs from the changes caused by the HF-CD diet.

## 6. Conclusions

The feeding pigs for tree month with high fat-carbohydrates diet is reflected in the physical properties of RBCs. Such RBCs are characterized by a lower osmotic fragility, a slightly reduced transmembrane potential, and a changed shape in comparison to RBCs feed with normal diet. Thus, the results indicate that changes in physical properties of RBCs may be useful, along with other research, in the diagnosis of obesity-related diseases. Furthermore, the research has shown, that phenolic compounds contained in *Cistus incanus* L. not only possess high biological activity but can also slightly reverse HF-CD-induced changes in RBCs. Their presence in the membrane induces changes in osmotic resistance, transmembrane potential, and the shape of RBCs in in vitro conditions. They bind to the outer monolayer of RBCs membrane and mainly interact with polar heads of lipids in the water-lipid interface; therefore, they effectively protect RBCs against oxidative damage.

## Figures and Tables

**Figure 1 materials-14-01050-f001:**
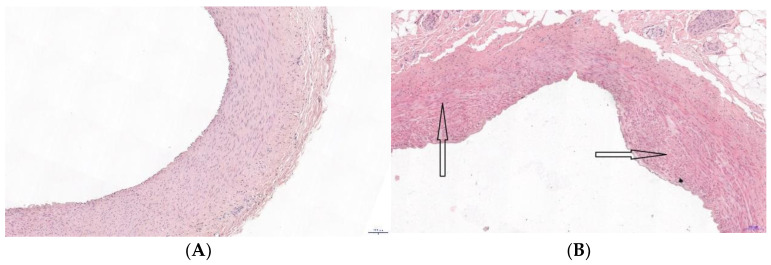
Histopathological examination of left descendens coronary artery. Hematoxylin-eosin staining of samples of heart tissue of swine (**A**) control group, (**B**) arteriosclerotic lesion (black arrow) in a pig with the representative remodeling- group fed a high fat-carbohydrates diet (HF-CD).

**Figure 2 materials-14-01050-f002:**
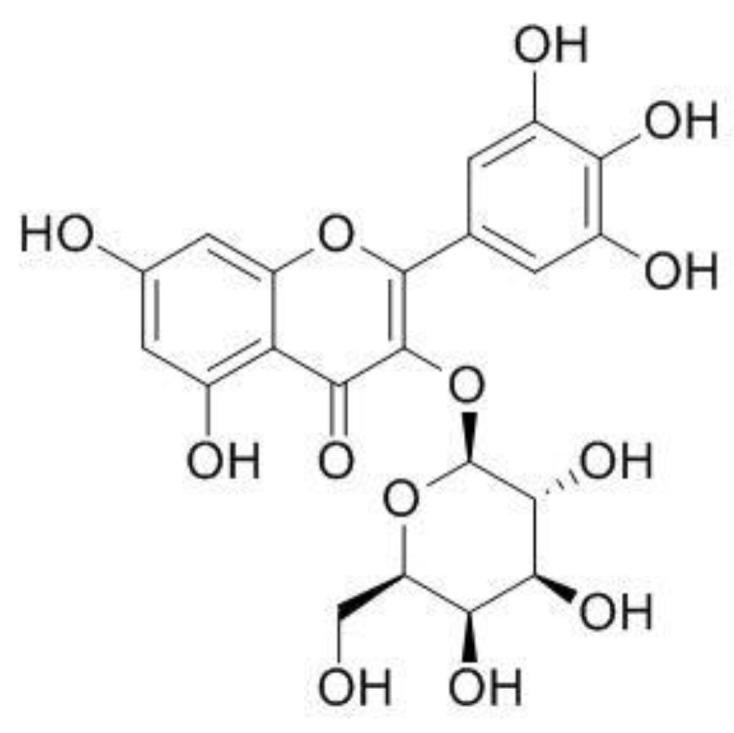
The chemical structure of myricetin 3-rhamnoside (M3R).

**Figure 3 materials-14-01050-f003:**
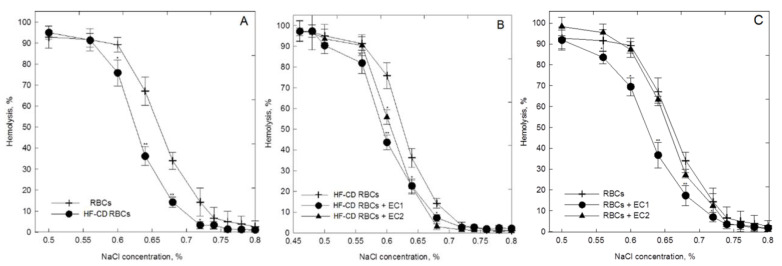
Percentage of hemolysis of control erythrocytes (RBCs) and these modified by 0.05 mg/mL of extracts form *Cistus incanus* L. (EC1 and EC2) in hypotonic and isotonic solutions of sodium chloride Panel **A**, **B**, **C**. The RBCs were obtained from healthy pig and pig fed a high fat-carbohydrates diet (HF-CD).

**Figure 4 materials-14-01050-f004:**
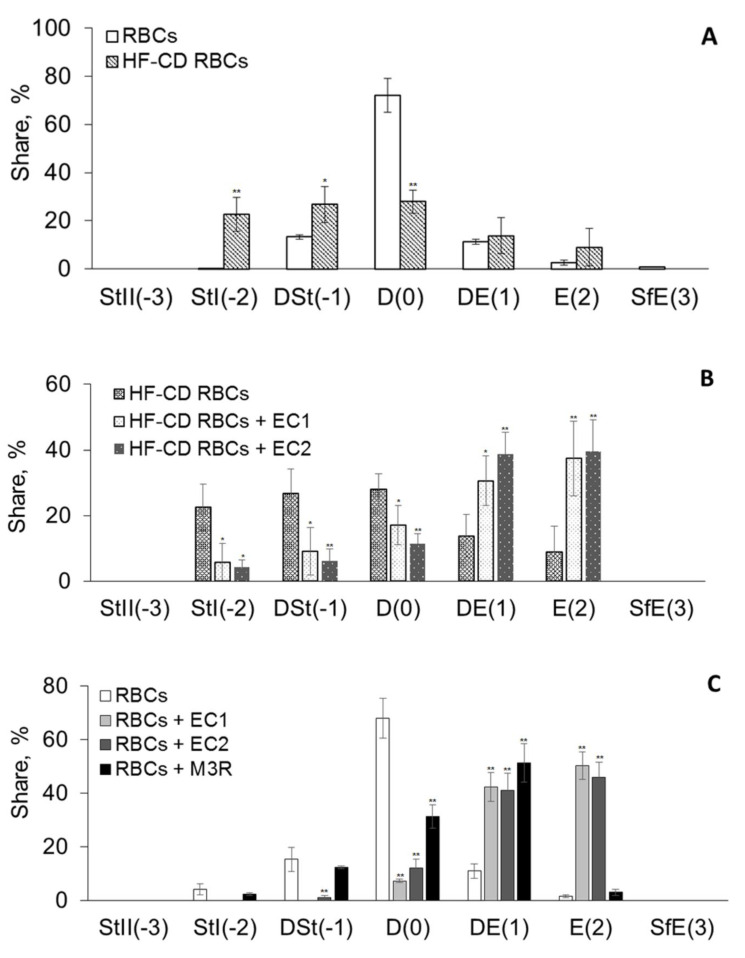
Percentage share of erythrocytes shapes obtained from control (RBCs) (Panel **A**) * and pigs fed high fat-carbohydrates diet (HF-CD RBCs) (Panel **B**) ** and these modified by EC1, EC2 and M3R used at 0.05 mg/mL (Panel **C**) ***.

**Table 1 materials-14-01050-t001:** The selected results of hematological and biochemical blood tests of control pigs and pigs fed for 3 months of HF-CD [1]. The results are presented as an average ± SD.

Parameter	Control *n* = 11	HF-CD *n* = 9
RBC 10^3^/L	7.54 ± 0.36	6.21 ± 0.89
Hb, mmol/L	6.27 ± 0.53	7.54 ± 0.36
Ht L/L	0.38 ± 0.02	0.34 ± 0.03
MCV, fL	50.7 ± 3.80	55.1 ± 4.06
MCH, fmoL	0.92 ± 0.07	1.02 ± 0.08
MCHC, mmol/L	16.52 ± 5.45	18.5 ± 0.32
Cholesterol, mmol/L	2.45 ± 0.25	2.78 ± 0.28↑
HDL, mmol/L	1.40 ± 0.09	1.23 ± 0.07↓
LDL, mmol/L	1.07 ± 0.18	1.36 ± 0.16↑
TGC, mmol/L	0.31 ± 0.19	0.67 ± 0.27↑
Glucose, mmol/L	5.40 ± 0.19	6.25 ± 3.01↑

**Table 2 materials-14-01050-t002:** Percentage contained extracts from *Cistus incanus* L. (EC1 and EC2).

No.	Compound	EC1	EC2
1	gaolloyl glucose	0.09	0.00
2	gallic acid	0.00	0.68
3	punicalin isomer	0.24	0.00
4	gallocatechin trimer	0.87	0.39
5	gallocatechin dimer	0.71	1.35
6	gallocatechin-(4α-8)-catechin	1.25	0.40
7	digalloyl glucose	0.00	0.04
8	gallocatechin	0.00	3.66
9	punicalagin isomer	1.21	0.47
10	cornusiin B	0.75	0.23
11	bis-HHDP-glucose	0.24	1.22
12	prodelphinidin dimer	0.83	0.46
13	HHDP-digalloyl-glucoside	0.00	0.44
14	(-)epicatechin	1.24	3.07
15	galloyl-HHDP-glucoside	0.00	0.06
16	punicalagin gallate	0.34	0.00
17	galloylprodelphinidin trimer	0.00	0.28
18	HHDP-hex	0.21	0.00
19	galloyl-prodelphinidin trimmer	0.00	0.32
20	digalloyl-HDDP-glucoside(pedunculagin II)	0.00	0.57
21	myricetin-3-O-galactoside	3.44	5.72
22	myricetin-3-O-glucoside	0.38	0.70
23	myricetin-O-xyloside	1.97	7.10
24	ellagic acid rutinoside	0.22	0.00
25	myricetin 3-rhamnoside	7.71	47.30
26	quercetin-3-O-galactoside	0.57	2.04
27	quercetin-3-O-glucoside	0.13	0.43
28	myricetin-pentoside	0.23	0.00
29	quercetin-pentoside	0.43	1.94
30	kaempferoldimethyletherhexoside	0.10	0.59
31	kaempferol-3-O-glucoside	0.13	0.61
32	quercetin-3-O-rhamnoside	0.61	4.73
33	myricetin -rutinoside	0.08	0.85
34	myricetin -rhamno-glucoside	0.02	0.14
35	quercetin rhamno-glucoside	0.01	0.14
36	kaempferol -O-rutinoside	0.07	0.59
37	kaempferol rhamno-glucoside	0.02	0.18
	**Total**	**24.11**	**87.15**

**Table 3 materials-14-01050-t003:** Values of transmembrane potential calculated for control erythrocytes (RBCs) and these modified by 0.05 mg/mL of extracts form *Cistus incanus* L. (EC1 and EC2). RBCs collected from healthy (RBCs) and fed a high fat-carbohydrates diet (HF-CD RBCs) pigs.

Sample	Membrane Potential, mV
RBCs	−13.72 ± 2.58
RBCs + EC1	−17.19 ± 3.95 *
RBCs + EC2	−19.00 ± 2.63 *
RBCs + M3R	−16.9 ± 3.2
HF-CD RBCs	−8.79 ± 2.30 **
HF-CD RBCs + EC1	−9.19 ± 1.38
HF-CD RBCs + EC2	−12.76 ± 1.77

* statistically significant differences between the control RBCs and extracts-treated RBCs are marked (α = 0.05); ** statistically significant differences between the HF-CD and control RBCs are marked (α = 0.1), statistical analysis was conducted using the Dunnett test.

**Table 4 materials-14-01050-t004:** Concentration of EC1, EC2 and M3R responsible for 50% inhibition.

Sample	Concentration RBCs, µg/mL
EC1	5.94 ± 0.35
EC2	5.49 ± 0.49
M3R	7.71 ± 0.17

## Data Availability

On request of those interested.

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
