# Peer review of "Comparison of Osmotic Resistance, Shape and Transmembrane Potential of Erythrocytes Collected from Healthy and Fed with High Fat-Carbohydrates Diet (HF-CD) Pigs—Protective Effect of Cistus incanus L. Extracts"

_materials, 2021, doi:10.3390/ma14041050_

Round 1

Reviewer 1 Report

            The manuscript by S. Cyboran-Mikołajczyk and colleagues addresses the effect of high fat-carbohydrates diet on vital properties of pig RBC. The properties are characterized by several methods that complement each other. Besides, two extracts from Cistus Incanus are analyzed in view of their potentially protective action on RBCs. The authors suggest an important role of membrane composition and lipid distribution in proper function of RBCs, and point to possibility to use RBC state as a marker of obesity-related diseases.

I have several critical comments.

1) I doubt that the essence of the manuscript fits the scope of the journal “Materials”. It is difficult to relate the manuscript with any of “Topics Covered” listed at the journal website (https://www.mdpi.com/journal/materials/about). The question “Why the article will be interesting to the readers of “Materials”?” should be addressed in the manuscript.

2) The manuscript is poorly written and organized.

Table 1 (page 4, lines 137-140) and Figure 1 (page 5), and the describing text represent the results of the work, and thus better fit the “Results” rather than “Materials and Methods” section.

Throughout the text, the authors name the plant, the extracts of which were analyzed in the work, either Citrus Incanus or Cistus Incanus. As far as I understand, the correct version is Cistus Incanus. However, “Citrus Incanus” stands even in the title of the manuscript, and in many places below (lines 34, 41, 43, 81, 99, 423).

Throughout the text, the authors name the diet either HF-CD or HFD (lines 73, 230, 257, 258, 265, 284, 289, 305, 329, 361). The latter abbreviation was not even introduced; I can only suppose that it corresponds to “High Fat Diet”. Do “HF-CD” and “HFD” mean the same diet? Why different abbreviations are used?

There are two “Table 1”-s in the text — on page 4 and on page 6. Thus, the numeration of all tables in the manuscript is not correct.

The authors randomly use decimal point “.” (e.g., Table 1, page 4) and decimal comma “,” (e.g., line 258) in the text.

There are numerous misprints throughout the text (e.g., line 113, 337).

3) Table 1 (page 6). The 3rd and the 4th column heads are “CE1” and “CE2”. These abbreviations were not introduced. Probably, “EC1” and “EC2” should stand instead.

Besides, compounds #3 and #4 have the same name (punicalin isomer), although their content in the extract is different. The same concerns compounds #10, #12, #14 (gallocatechin dimer), as well as compounds #17, #18 (bis-HHDP-glucose) and #26, #29 (galloylprodelphinidin trimer / galloyl-prodelphinidin trimmer). If the compound #30 should read “digalloyl-HDDP-glucoside(pedunculagin II)” instead of “gigalloyl-HDDP-glucoside(pedunculagin II)”, then this compound coincides with the compound #22 (HHDP-digalloyl-glucoside). The different content of the same compound in the extract should be explained. If the compounds are actually different, the difference should be reflected in their names in the Table.

7) Line 199: “… (0.5 - 0.86 %) NaCl solutions …”.

What kind of percent is used here (weight, volume, molar, etc.)?

8) Lines 222-223: “… the external potassium concertation (K+EX), for which no change in DiSC3(5) florescence intensity occurs upon valinomycin addition, was calculated.

These data are not shown in the manuscript. It seems that they were not used at all; at least, their use is not described. What was the reason for gathering the data?

9) Lines 241, 243. Here the abbreviation AAPH used for the first time. It is not defined in the section “Abbreviations” and its definition appears much later (line 347, title of the Table 3). The abbreviation should be defined near line 241.

10) The authors used the word “compound(s)” (lines 244, 246, 337, 431); the exact compound meant is not quite clear. The compound(s) should be explicitly named in all these cases.

11) Lines 257-260.

The procedure of determination of hemolysis degree of RBC and HF-CD RBC is not described in the section “Materials and Methods”. There is only the description of the procedure of determination of hemolytic activity of extracts, and not of hemolysis of unmodified RBCs. What are the reasons of hemolysis of RBCs non-treated with extracts?

12) Lines 298-299: “The calculated results presented in Table 2 indicate that the transmembrane potential of HF-CD RBCs negligible differ from the transmembrane potential of RBCs collected from healthy pigs.

The presented values of the transmembrane potential are reliably different (rather than negligibly); they do not intersect within the confidence intervals (–13.72 ± 2.58 and –8.79 ± 2.30 mV). Thus, the difference is not negligible.

13) Lines 375-378 (“The spectroscopic studies do not show increased hemolysis of RBCs obtained from HF-CD fed pigs, but they showed their increased resistance to changes in osmotic pressure. This is indicated by a shift in the hemolytic curve towards lower NaCl concentrations in comparison to control RBCs.”) and lines 432-434 (“In the case of control RBCs, EC1 and EC2 reduce osmotic fragility, making the membrane more resistant to changes in osmotic pressure of the surrounding environment.”)

The statement that the diet or extracts make the RBC membrane more resistant to the osmotic pressure is quite speculative. The mechanical details of the effect of the osmotic stress on the membrane were not explicitly studied in the work. The effect was only characterized by the relative amount of leaked hemoglobin. However, the data obtained can be interpreted in the exactly opposite way: that the diet and the extracts make the RBC membrane more fragile and thus less resistive to the osmotic stress. Indeed, for example, in response to the osmotic stress, numerous small pores can form in the fragile membrane just because of its high fragility. These pores are too small for hemoglobin to pass; however, multiple small pores allow rapid adjustment of the internal concentration of the salt to effectively nullify the osmotic gradient. After the gradient relaxation, the pores rapidly seal, in accordance with the classic theory of poration by Derjaguin & Gutop [Derjaguin, B. V., and Gutop, Yu. V., Kolloid. zh. 24, 431 (1962); Dokl. AN SSSR 153, 859 (1963)]. Thus, the hemoglobin leakage through the fragile membrane may be zero. On the contrary, if unmodified RBC membrane is less fragile, then a single large pore may form in response to the osmotic stress. This pore is large enough to allow hemoglobin leakage. Thus, without explicit check and proof, the statements made in the manuscript look ungrounded.

A possible way to judge on the increased/decreased stability of RBC membrane is to quantitatively determine the cholesterol content of unmodified vs. modified membranes. In physiological range of concentrations, cholesterol should definitely increase the membrane stability with respect to pore formation and decrease the water permeability of the membrane. Other way is to take unmodified RBCs, and artificially increase the level of cholesterol in the plasma membranes by treating the cells with beta-methyl-cyclodextrin loaded with cholesterol; then compare the hemolysis of unmodified and treated cells and relate with the hemolysis of HF-CD RBCs.

14) Lines 380-382: “However, there is a clear downward trend in the membrane potential of HF-CD RBCs (less negative potential value) and the associated slight increase in extracellular potassium ion concentration.

The data on variation of “extracellular potassium ion concentration” are not presented in the manuscript. Thus, the statement looks ungrounded.

15) Line 387, 400, 402-403, 412. The authors use the phrase “(class) distribution of (phospho)lipids”.

It is not clear, what kind of distribution is meant. I have at least two hypotheses: 1) distribution of lipids between leaflets (inner and outer) of plasma membrane; 2) lateral distribution of lipids within a membrane leaflet, resulting in formation of, e.g., ordered domains (rafts). This issue should be clarified.

16) Lines 406-408: “The presence of additional cholesterol molecules in RBC membrane may lead to an expansion of outer leaflet relative to the inner leaflet and result in the formation of characteristic irregular shape of the cholesterol enriched RBCs.

The characteristic time of cholesterol flip-flop (exchange of residentiary leaflet of the membrane) is of the order of 1 second. Thus, on characteristic timescales of the HF-CD diet (3 months), cholesterol cannot “lead to an expansion of outer leaflet relative to the inner leaflet”. Moreover, in the membranes of composition resembling that of the outer leaflet of a plasma membrane, in the certain range of concentrations, the addition of cholesterol leads to decrease of the total area of the membrane, due to condensing effect of cholesterol on lipid chains. Thus, the proposed reason of “formation of characteristic irregular shape” looks ungrounded.

17) Lines 439-441: “Observed changes in osmotic stability, shape and transmembrane potential of RBCs are probably the results of binding of the extracts components mainly to the hydrophilic part of RBC membrane.

It is not clear how the part of the membrane of preferential binding of the extract components was determined — this question was not experimentally addressed in the manuscript. Polyphenols are amphiphilic, and thus they should bind both to hydrophilic and hydrophobic parts of the membrane, i.e., most probably, reside at the polar/hydrophobic interface of a lipid monolayer.

18) Lines 468-470: “These results indicate, that diet enriched with Cistus incanus L., can protect RBCs from the changes caused by the HF-CD diet.

This statement is completely ungrounded. The results can only indicate that the injection of Cistus incanus L. extracts (e.g., into the blood stream) may modify RBC membranes to partially compensate the effect of HF-CD diet. The effect of the “diet enriched with Cistus incanus L.” was not studied in the work, as only extracts were applied directly to RBCs; there were no “diet enriched with Cistus incanus L.” as pigs were not fed by Cistus incanus L. Moreover, the protective properties of the extracts were not established in the work, as the extracts were applied to RBCs only after the HF-CD diet. To establish the protective properties, the extracts should be applied before or in the course of the HF-CD diet, and they should (partially) compensate the changes caused by the HF-CD diet. This was not done/observed in the work

Author Response

RESPOND TO REVIEWERS COMMENTS:

Thank you very much for valuable suggestions that have contributed to the improvement of the quality of work.

#Reviewer 1

The manuscript by S. Cyboran-Mikołajczyk and colleagues addresses the effect of high fat-carbohydrates diet on vital properties of pig RBC. The properties are characterized by several methods that complement each other. Besides, two extracts from Cistus Incanus are analyzed in view of their potentially protective action on RBCs. The authors suggest an important role of membrane composition and lipid distribution in proper function of RBCs, and point to possibility to use RBC state as a marker of obesity-related diseases. I have several critical comments.

1) I doubt that the essence of the manuscript fits the scope of the journal “Materials”. It is difficult to relate the manuscript with any of “Topics Covered” listed at the journal website (https://www.mdpi.com/journal/materials/about). The question “Why the article will be interesting to the readers of “Materials”?” should be addressed in the manuscript.

 Ad. 1 In this work, the basic research on the biological activity of biological material, i.e. two extracts of Cistus Incantus L. was carried out. In particular, it was determined the effect of the extracts on the physical properties of erythrocytes collected from pigs fed with a balanced diet and a diet enriched in carbohydrates and saturated fat (HF-CD).

2) The manuscript is poorly written and organized.

Table 1 (page 4, lines 137-140) and Figure 1 (page 5), and the describing text represent the results of the work, and thus better fit the “Results” rather than “Materials and Methods” section.

Ad. 2.1 The results presented  in Table 1 and Figure 1 are intended to confirm that the 3-month HF-CD diet is not indifferent to the pig’s organism and induces arterial changes. Used in this work swine model of atherosclerosis was described earlier in Ząbek et al., 2017. [page 3, line 111]. The aim of the present work was to determine the impact of HF-CD and two extracts of Cistus on physical properties of erythrocytes, therefore, the above-mentioned results are included in “Materials and methods'' section and not in the “Results'' section.

Throughout the text, the authors name the plant, the extracts of which were analyzed in the work, either Citrus Incanus or Cistus Incanus. As far as I understand, the correct version is Cistus Incanus. However, “Citrus Incanus” stands even in the title of the manuscript, and in many places below (lines 34, 41, 43, 81, 99, 423).

Ad. 2.2. In this work two extracts of Cistus Incantus L. obtained using different extraction procedures were studied. All errors in the Latin name of the compound have been corrected throughout the manuscript (page 1, title and lines 36; page 2, lines 43 and 46; page 3, lines 84 and 103, page 18, line 479) .

Throughout the text, the authors name the diet either HF-CD or HFD (lines 73, 230, 257, 258, 265, 284, 289, 305, 329, 361). The latter abbreviation was not even introduced; I can only suppose that it corresponds to “High Fat Diet”. Do “HF-CD” and “HFD” mean the same diet? Why are different abbreviations used?

Ad. 2.3. In this study pigs were fed a balanced diet and a diet rich in carbohydrates and saturated fatty acids (HF-CD) were used. The abbreviation HFD, as reviewer correctly pointed out, indicates a high-fat diet and its use within the manuscript results from the omissions of the authors. All HFD abbreviations (except for line 73), have been removed and revised on HF-CD (page 4, table 1; page 10, lines 248, 273 and 274; page 11, lines: 281, 283, 295, 297 and 301; page 12, lines: 306, 323 and 328; page 13, line 349; page 14, line 381; page 18, line 503).

There are two “Table 1”-s in the text — on page 4 and on page 6. Thus, the numeration of all tables in the manuscript is not correct.

Ad. 2.4 The authors corrected the numbering of tables throughout the work.

The authors randomly use decimal point “.” (e.g., Table 1, page 4) and decimal comma “,” (e.g., line 258) in the text.

Ad. 2.5. The authors standardized the decimal notation  i.e. removed the decimal comma “;” and used decimal point “.” throughout the manuscript (page 10, lines 274-275).

There are numerous misprints throughout the text (e.g., line 113, 337).

Ad. 2.6 The authors critically reviewed the manuscript trying to correct all typographical errors.

3) Table 1 (page 6). The 3rd and the 4th column heads are “CE1” and “CE2”. These abbreviations were not introduced. Probably, “EC1” and “EC2” should stand instead.

Ad.3.1 It was the mistake in the abbreviations of the extract. The correct forms are EC1 and EC2 and they were used in Table 2.

Besides, compounds #3 and #4 have the same name (punicalin isomer), although their content in the extract is different. The same concerns compounds #10, #12, #14 (gallocatechin dimer), as well as compounds #17, #18 (bis-HHDP-glucose) and #26, #29 (galloylprodelphinidin trimmer / galloyl-prodelphinidin trimmer). If the compound #30 should read “digalloyl-HDDP-glucoside(pedunculagin II)” instead of “gigalloyl-HDDP-glucoside(pedunculagin II)”, then this compound coincides with the compound #22 (HHDP-digalloyl-glucoside). The different content of the same compound in the extract should be explained. If the compounds are actually different, the difference should be reflected in their names in the Table.

Ad.3.2.The authors mistakenly added the old version of table 2. This version should contain additional column no 2, that contains the retention time of the compounds. Therefore, compounds of the same name, e.g. procyanidin dimer from lines 6,10,12,14 differ only in retention time. In the revised version of the manuscript the amount of the compounds belonging to the same group i.e. procyanidin dimer have been summed (Table 2).

4) Line 199: “… (0.5 - 0.86 %) NaCl solutions …”.

What kind of percent is used here (weight, volume, molar, etc.)?

Ad.4. In osmotic resistance tests the erythrocytes were suspended in different solutions of NaCl. The solutions were prepared by dissolving the appropriate mass of NaCl in the distilled water (weight percent).This explanation has been added to the manuscript. 

5) Lines 222-223: “… the external potassium concertation (K+EX), for which no change in DiSC3(5) florescence intensity occurs upon valinomycin addition, was calculated.

These data are not shown in the manuscript. It seems that they were not used at all; at least, their use is not described. What was the reason for gathering the data?

Ad.5 The external concentration of K+ex is necessary to determine the transmembrane potential of erythrocytes (Equation). The values of K+ex obtained from the plots were used to determine the transmembrane potential of erythrocytes. As the transmembrane potential depends mainly on this concentration, the authors of the study have only included the final results.

6) Lines 241, 243. Here the abbreviation AAPH used for the first time. It is not defined in the section “Abbreviations” and its definition appears much later (line 347, title of the Table 3). The abbreviation should be defined near line 241.

Ad.6 The AAPH abbreviation was defined for the first time in “Reagents” ( page 8, line 186). The authors additionally explained this abbreviation in place where it was used for the first time (page 10, line 261).

7) The authors used the word “compound(s)” (lines 244, 246, 337, 431); the exact compound meant is not quite clear. The compound(s) should be explicitly named in all these cases.

Ad. 7. The authors of the work listed the specific names of  chemical compounds, in lines indicated by the reviewer.

8) Lines 257-260.

The procedure of determination of hemolysis degree of RBC and HF-CD RBC is not described in the section “Materials and Methods”. There is only the description of the procedure of determination of hemolytic activity of extracts, and not of hemolysis of unmodified RBCs. What are the reasons of hemolysis of RBCs non-treated with extracts?

Ad. 8 The aim of the hemolytic studies was to determine the effect of HF-CD on the degree of hemolysis of RBCs. The comparison of the level of hemolysis of RBCs obtained from pigs fed a balanced diet and HF-CD, did not show an increased number of hemolyzed blood cells. The procedure of this study was similar to that of extracts hemolytic activity, but phosphate buffer was added instead of extracts. The authors modified the procedure of  hemolytic activity of extracts and added it under the name of “the degree of hemolysis of RBCs” into the Method section (page 8, lines 192-206).

9) Lines 298-299: “The calculated results presented in Table 2 indicate that the transmembrane potential of HF-CD RBCs negligible differ from the transmembrane potential of RBCs collected from healthy pigs.

The presented values of the transmembrane potential are reliably different (rather than negligibly); they do not intersect within the confidence intervals (–13.72 ± 2.58 and –8.79 ± 2.30 mV). Thus, the difference is not negligible.

 Ad. 9 According to the reviewer suggestion that calculated for RBCs and HF-CD RBCs confidence intervals of transmembrane potencjal do not intersect ( -6,49:-11.09 and -11.14: 16.3), authors changed the conclusion, indicating a statistically significant slight increase in the transmembrane potential of the RBCs obtained from pigs fed a HF-CD (page 12, lines 315-318, table 3).

10) Lines 375-378 (“The spectroscopic studies do not show increased hemolysis of RBCs obtained from HF-CD fed pigs, but they showed their increased resistance to changes in osmotic pressure. This is indicated by a shift in the hemolytic curve towards lower NaCl concentrations in comparison to control RBCs.”) and lines 432-434 (“In the case of control RBCs, EC1 and EC2 reduce osmotic fragility, making the membrane more resistant to changes in osmotic pressure of the surrounding environment.”)

The statement that the diet or extracts make the RBC membrane more resistant to the osmotic pressure is quite speculative. The mechanical details of the effect of the osmotic stress on the membrane were not explicitly studied in the work. The effect was only characterized by the relative amount of leaked hemoglobin. However, the data obtained can be interpreted in the exactly opposite way: that the diet and the extracts make the RBC membrane more fragile and thus less resistive to the osmotic stress. Indeed, for example, in response to the osmotic stress, numerous small pores can form in the fragile membrane just because of its high fragility. These pores are too small for hemoglobin to pass; however, multiple small pores allow rapid adjustment of the internal concentration of the salt to effectively nullify the osmotic gradient. After the gradient relaxation, the pores rapidly seal, in accordance with the classic theory of poration by Derjaguin & Gutop [Derjaguin, B. V., and Gutop, Yu. V., Kolloid. zh. 24, 431 (1962); Dokl. AN SSSR 153, 859 (1963)]. Thus, the hemoglobin leakage through the fragile membrane may be zero. On the contrary, if unmodified RBC membrane is less fragile, then a single large pore may form in response to the osmotic stress. This pore is large enough to allow hemoglobin leakage. Thus, without explicit check and proof, the statements made in the manuscript look ungrounded.

A possible way to judge on the increased/decreased stability of RBC membrane is to quantitatively determine the cholesterol content of unmodified vs. modified membranes. In physiological range of concentrations, cholesterol should definitely increase the membrane stability with respect to pore formation and decrease the water permeability of the membrane. Other way is to take unmodified RBCs, and artificially increase the level of cholesterol in the plasma membranes by treating the cells with beta-methyl-cyclodextrin loaded with cholesterol; then compare the hemolysis of unmodified and treated cells and relate with the hemolysis of HF-CD RBCs.

 Ad. 10 The authors fully agree with the receiver that  the mechanical details of the effect of the osmotic stress on the membrane were not explicitly studied in the work. Therefore, the authors of the study refined the conclusions by pointing out that the increase in resistance is understood as a reduction in hemoglobin outflow. Moreover, on the basis of the conducted research, the authors are not able to explain the exact mechanism responsible for the observed effect in osmotic resistance. Furthermore, at the moment, the authors are not able to carry out additional research due to the lack of the access to RBCs, but in the future, they will take into account the reviewer's suggestions when planning the next research (page 15, lines 399-400; page 17, 460).

14) Lines 380-382: “However, there is a clear downward trend in the membrane potential of HF-CD RBCs (less negative potential value) and the associated slight increase in extracellular potassium ion concentration.

The data on variation of “extracellular potassium ion concentration” are not presented in the manuscript. Thus, the statement looks ungrounded.

Ad 14. According to the used model, the transmembrane potential depends on the ratio of external/inertial K+ concentration, so its increase must be associated with a slight increase in extracellular potassium ion concentration (assuming that intracellular K+ concentration do not change).

15) Line 387, 400, 402-403, 412. The authors use the phrase “(class) distribution of (phospho)lipids”.

It is not clear what kind of distribution is meant. I have at least two hypotheses: 1) distribution of lipids between leaflets (inner and outer) of plasma membrane; 2) lateral distribution of lipids within a membrane leaflet, resulting in formation of, e.g., ordered domains (rafts). This issue should be clarified.

Ad. 15 The authors specified what distribution of lipids they meant (page 16, lines 410, 423-424 and 437). 

16) Lines 406-408: “The presence of additional cholesterol molecules in RBC membrane may lead to an expansion of outer leaflet relative to the inner leaflet and result in the formation of characteristic irregular shape of the cholesterol enriched RBCs.

The characteristic time of cholesterol flip-flop (exchange of residentiary leaflet of the membrane) is of the order of 1 second. Thus, on characteristic timescales of the HF-CD diet (3 months), cholesterol cannot “lead to an expansion of outer leaflet relative to the inner leaflet”. Moreover, in the membranes of composition resembling that of the outer leaflet of a plasma membrane, in the certain range of concentrations, the addition of cholesterol leads to decrease of the total area of the membrane, due to condensing effect of cholesterol on lipid chains. Thus, the proposed reason of “formation of characteristic irregular shape” looks ungrounded.

Ad.16 The authors clarify their conclusions, indicating that the presence of additional cholesterol particles affect the physical properties of RBCs, mainly fluidity of the membrane, and this in turn may affect  the shape of the blood cell.

17) Lines 439-441: “Observed changes in osmotic stability, shape and transmembrane potential of RBCs are probably the results of binding of the extracts components mainly to the hydrophilic part of RBC membrane.

It is not clear how the part of the membrane of preferential binding of the extract components was determined — this question was not experimentally addressed in the manuscript. Polyphenols are amphiphilic, and thus they should bind both to hydrophilic and hydrophobic parts of the membrane, i.e., most probably, reside at the polar/hydrophobic interface of a lipid monolayer.

Ad.17 Our ealier studies using fluorimetric methods showed that the polyphenols contained in the EC1 and EC2 increase the lipid polar head motility in the hydrophilic part of the erythrocyte membrane and slightly increase fluidity of the membrane. The observed changes in the physical properties of the membrane indicate that they are located mainly in the hydrophilic membrane area, forming a barrier on the surface to protect the membrane from oxidative damage. [Cyboran Mikołajczyk S.,  Bonarska-Kujawa D., Męczarska K., Włoch A., Oszmiański J., Pasławski R., Kleszczyńska H.  Is it possible to use polyphenol compounds contained in Cistus incanus L. in cancer prevention? Chapter in “Herbal plants, natural cosmetics and functional food”, 2017, pp.357-369.]. Taking into account the size and chemical structure of the extracts ingredients and the results of fluorimetric studies, it may be concluded that the slight changes in the hydrophobic area of the membrane are the results of changes in the hydrophilic. Extract’s ingredients are rather not able to penetrate into the region of the hydrocarbon chains, but they may electrostatically interact with the polar head of lipids or form hydrogen bonds with their OH groups. The authors added the explanation and the citation into the manuscript (page 18, lines: 487-495; page 23, lines: 658-661) 

18) Lines 468-470: “These results indicate, that diet enriched with Cistus incanus L., can protect RBCs from the changes caused by the HF-CD diet.

This statement is completely ungrounded. The results can only indicate that the injection of Cistus incanus L. extracts (e.g., into the blood stream) may modify RBC membranes to partially compensate the effect of HF-CD diet. The effect of the “diet enriched with Cistus incanus L.” was not studied in the work, as only extracts were applied directly to RBCs; there were no “diet enriched with Cistus incanus L.” as pigs were not fed by Cistus incanus L. Moreover, the protective properties of the extracts were not established in the work, as the extracts were applied to RBCs only after the HF-CD diet. To establish the protective properties, the extracts should be applied before or in the course of the HF-CD diet, and they should (partially) compensate the changes caused by the HF-CD diet. This was not done/observed in the work

Ad.18 The authors agree with the reviewer, therefore, on the basis of the above comments, this statement was corrected and rewritten by the authors (pp. 19, lines 505-508).

Reviewer 2 Report

The authors described membrane physicochemical properties of RBCs obtained from an animal model fed HF-CD. Furthermore, the authors examined the effect of the cistus extracts (EC1 and EC2) to RBCs for the stability to the osmotic pressures, alteration of the membrane potentials, and protection against oxidation. The data shown in this manuscript will be a good initial point to launch studies of HF-CD RBCs in terms of the diagnostic, clinical and mechanism study in the future. The reviewer thought comprehensive discussion for the data and surrounding evidences are also useful for the cross-discipline researchers. The reviewer raise following points for further discussion.

  1. The reviewer intrigues about the effect of cholesterol and other lipid components in RBC membranes in terms of the resistance to the osmotic pressure. The authors discussed in page 14 around line 385 “The observed changes in the physical properties of the membrane are probably caused by impaired lipid composition.”. The following sentences particularly discussed the effect of cholesterol to the RBC membranes. The reviewer has interest whether the resistance of HF-CD RBCs changes to be comparable to the normal RBCs if the membrane cholesterol was extracted by methyl-beta-cyclodextrin?

  1. The reviewer has interest for the membrane effect of the EC1 and 2. The structures of EC1 and 2 shown in Table 1 mostly have hydrophilic sugar headgroups with hydrophobic aglycone structures, like mild detergents. The authors described page 16 line around 459, “Their location on membrane-water interface,….”. When the EC1 or 2 affects RBC membranes, the hydrophobic core of these compound were inserted into membrane and directly decrease the membrane stability? Can these molecules directly associate with cholesterol in membrane or serum to assist the clearance like bile acids and some saponins?

  1. The reviewer has a question about the concentration of EC1, EC2, or M3R in a series of the experiments. The reviewer concerned it may be difficult to directly compare the activity of EC1 and EC2 because they are the complex mixtures, and thus the results need to be interpreted the data carefully. Could the authors found concentration dependency of EC1, EC2, or M3R for their activities? If the authors had carried out their experiments in a single concentration, it is necessary to explain why used the specific concentrations? Is it related to CMC, or any other reasons?

Minor point

  1. Please show the chemical structures of M3R and some of the typical compounds found in EC1 and 2 in Table 1 or a figure.

Author Response

RESPOND TO REVIEWERS COMMENTS:

Thank you very much for valuable suggestions that have contributed to the improvement of the quality of work.

#Reviewer 2

The authors described membrane physicochemical properties of RBCs obtained from an animal model fed HF-CD. Furthermore, the authors examined the effect of the cistus extracts (EC1 and EC2) to RBCs for the stability to the osmotic pressures, alteration of the membrane potentials, and protection against oxidation. The data shown in this manuscript will be a good initial point to launch studies of HF-CD RBCs in terms of the diagnostic, clinical and mechanism study in the future. The reviewer thought comprehensive discussion for the data and surrounding evidences are also useful for the cross-discipline researchers. The reviewer raise following points for further discussion.

  1. The reviewer intrigues about the effect of cholesterol and other lipid components in RBC membranes in terms of the resistance to the osmotic pressure. The authors discussed in page 14 around line 385 “The observed changes in the physical properties of the membrane are probably caused by impaired lipid composition.”. The following sentences particularly discussed the effect of cholesterol to the RBC membranes. The reviewer has interest whether the resistance of HF-CD RBCs changes to be comparable to the normal RBCs if the membrane cholesterol was extracted by methyl-beta-cyclodextrin?

Ad 1. The authors, on the basis of obtained results and literature data, indicate that the changes in the physical properties of HF-CD RBCs are probably associated with disturbed lipid distribution. It is highly likely that the cholesterol concentration in the membrane changes and hence the fluidity of the lipid bilayer. It is also highly likely that HF-CD may lead to local inflammation and oxidation of membrane components and in consequence to the increased amount of usaturated fatty acids. Literature data also indicate that the distribution of lipids belonging to different classes may vary [Havge et al.1991]), The authors believe that the changes observed in HF-CD RBCs are more advanced than just the increase of cholesterol level in the membrane. They think that the extraction of cholesterol from the membrane by using methyl-beta-cyclodextrin methods did not cause the properties of HF-CD membrane  will be comparable to those of control RBCs.

  1. The reviewer has interest for the membrane effect of the EC1 and 2. The structures of EC1 and 2 shown in Table 1 mostly have hydrophilic sugar headgroups with hydrophobic aglycone structures, like mild detergents. The authors described page 16 line around 459, “Their location on membrane-water interface,….”. When the EC1 or 2 affects RBC membranes, the hydrophobic core of these compound were inserted into membrane and directly decrease the membrane stability? Can these molecules directly associate with cholesterol in membrane or serum to assist the clearance like bile acids and some saponins?

Ad. 2 Our ealier studies using fluorimetric methods showed that the polyphenols contained in the EC1 and EC2 increase the lipid polar head motility in the hydrophilic part of the membrane and slightly increase fluidity of the erythrocyte membrane. The observed changes in the physical properties of the membrane indicate that they are located mainly in the hydrophilic membrane area, forming a barrier on the surface to protect the membrane from oxidative damage [Cyboran Mikołajczyk S.,  Bonarska-Kujawa D., Męczarska K., Włoch A., Oszmiański J., Pasławski R., Kleszczyńska H.  Is it possible to use polyphenol compounds contained in Cistus incanus L. in cancer prevention? Chapter in “Herbal plants, natural cosmetics and functional food”, 2017, pp.357-369]. Taking into account the size and chemical structure of the extract ingredients and the results of fluorimetric studies, it may be concluded that the slight changes in the hydrophobic area of the membrane are the results of changes in the hydrophilic area. The compounds are rather not able to penetrate into the region of the hydrocarbon chains, but they may electrostatically interact or form the hydrogen bonds with the OH groups of membrane lipids. The authors added the explanation and the citation into the manuscript (page 18, lines: 487-495; page 23, lines: 658-661)  

  1. The reviewer has a question about the concentration of EC1, EC2, or M3R in a series of the experiments. The reviewer concerned it may be difficult to directly compare the activity of EC1 and EC2 because they are the complex mixtures, and thus the results need to be interpreted the data carefully. Could the authors found concentration dependency of EC1, EC2, or M3R for their activities? If the authors had carried out their experiments in a single concentration, it is necessary to explain why used the specific concentrations? Is it related to CMC, or any other reasons?

Ad.3 The studies of the impact of EC1 and EC2 extracts on erythrocytes showed that the method of extraction of polyphenolic compounds from Cistus herb influences their biological activity. The use of the same concentrations allowed to indicate which of the preparations used has the greater ability to modify the membrane properties and thus the potential therapeutic properties. All experiments were carried out at least in two different concentrations (0.01 and 0.05 mg/mL). A concentration of 0.05 mg/ml was selected and used in order to show the difference in activity of both extracts. The studies have shown that ingredients of EC2 extract possess higher antioxidant potential (lower IC50 concentration), but their ability to membrane modification is not significantly different from that of EC1 components. The use of M3R the main component of both extracts at the same concentration (0,05mg/mL) indicates that M3R alone does not change the transmembrane potential of erythrocytes and induces the formation of only echinocytes. It means that biological activity of EC1 and EC2 only partially depends on their main component, whose content is 31% in EC1 and 54% in EC2 of all phenolic components, respectively. Extracts biological activity is much higher than activity of M3R, which indicates that the use of a mixture of Cistus polyphenols may have greater therapeutic and pro-health potential.

Minor point

Please show the chemical structures of M3R and some of the typical compounds found in EC1 and 2 in Table 1 or a figure.

The authors added the chemical structure of M3R - pages 7-8, Fig. 2. 

Reviewer 3 Report

Some more experiments should be made including membrane capacitance

Author Response

RESPOND TO REVIEWERS COMMENTS:

Thank you very much for valuable suggestions that have contributed to the improvement of the quality of work.

#Reviewer 3

Some more experiments should be made including membrane capacitance

In the described experiment, no membrane capacity tests were planned, but the valuable suggestion of the reviewer will be taken into account when planning further tests..

Round 2

Reviewer 1 Report

The authors satisfactory addressed all critical points.

Reviewer 2 Report

The authors carefully prepared the answers and  satisfied The reviewer.